# A Bioreactor-Based Yellow Fever Virus-like Particle Production Process with Integrated Process Analytical Technology Based on Transient Transfection

**DOI:** 10.3390/v15102013

**Published:** 2023-09-27

**Authors:** Gregor Dekevic, Tobias Tertel, Lars Tasto, Deborah Schmidt, Bernd Giebel, Peter Czermak, Denise Salzig

**Affiliations:** 1Institute of Bioprocess Engineering and Pharmaceutical Technology, University of Applied Sciences Mittelhessen, Wiesenstrasse 14, 35390 Giessen, Germany; gregor.dekevic@lse.thm.de (G.D.); lars.tasto@lse.thm.de (L.T.); deborah.schmidt@lse.thm.de (D.S.); peter.czermak@lse.thm.de (P.C.); 2Institute for Transfusion Medicine, University Hospital Essen, University of Duisburg-Essen, Virchowstrasse 179, 45147 Essen, Germany; tobias.tertel@uk-essen.de (T.T.); bernd.giebel@uk-essen.de (B.G.); 3Faculty of Biology and Chemistry, University of Giessen, Heinrich-Buff-Ring 17, 35392 Giessen, Germany

**Keywords:** stirred-tank bioreactor (STR), HEK 293T cells, lPEI, Process Analytical Technology (PAT), design of experiments, IFCM

## Abstract

Yellow Fever (YF) is a severe disease that, while preventable through vaccination, lacks rapid intervention options for those already infected. There is an urgent need for passive immunization techniques using YF-virus-like particles (YF-VLPs). To address this, we successfully established a bioreactor-based production process for YF-VLPs, leveraging transient transfection and integrating Process Analytical Technology. A cornerstone of this approach was the optimization of plasmid DNA (pDNA) production to a yield of 11 mg/L using design of experiments. Glucose, NaCl, yeast extract, and a phosphate buffer showed significant influence on specific pDNA yield. The preliminary work for VLP-production in bioreactor showed adjustments to the HEK cell density, the polyplex formation duration, and medium exchanges effectively elevated transfection efficiencies. The additive Pluronic F-68 was neutral in its effects, and anti-clumping agents (ACA) adversely affected the transfection process. Finally, we established the stirred-tank bioreactor process with integrated dielectric spectroscopy, which gave real-time insight in relevant process steps, e.g., cell growth, polyplex uptake, and harvest time. We confirmed the presence and integrity of YF-VLP via Western blot, imaging flow cytometry measurement, and transmission electron microscopy. The YF-VLP production process can serve as a platform to produce VLPs as passive immunizing agents against other neglected tropical diseases.

## 1. Introduction

Yellow Fever (YF) is a serious disease with 29,000–60,000 annual deaths [1]. It occurs in tropical and subtropical regions of Africa and South America, and is caused by a flavivirus, which is spread by the bite of mosquitos (primarily *Aedes aegypti*) [2,3]. Although a live-attenuated egg-derived vaccine against YF has been available since the 1930s [4], this vaccination alone cannot prevent YF outbreaks, as seen in Africa (2016) [5] and Brazil (2016–2017) [6]. A quick, easy, flexible, and inexpensive option is still missing to respond to such local epidemic YF outbreaks in the world.

One way of counteracting epidemic YF outbreaks is using virus-like particles. YF-VLPs can be applied in a passive immunization approach, as they are easily recognized by the immune system, triggering the production of anti-YF-virus-antibodies [7]. Sera containing anti-YF-virus-antibodies are then used to treat already-infected YF patients. VLPs are small, with ordered spheres consisting of viral structural proteins that spontaneously self-assemble into nanometer-sized particles when overexpressed [8,9,10]. VLPs are non-infectious, as they lack viral genetic material, but mimic the structure of authentic virus particles. To use YF-VLP as a passive immunization agent, a fast and efficient production must be established. There are many ways to produce VLPs using different expression systems, e.g., mammalian cells [11], insect cells [12], microbial cells [13], or plants [14]. Mammalian cells offer the decisive advantage of being able to produce important correct post-translational modification patterns, such as glycosylation. Especially, a transient VLP production process using mammalian, especially human cells, e.g., HEK 293T/17 SF suspension cells, offer a rapid and cost-effective platform to react to YF outbreaks. To deliver the negatively charged plasmid DNA (pDNA) through the negatively charged cell membrane into the cell interior, a cationic polymer can be used for transient transfection, e.g., linear polyethylenimine (lPEI). The cationic lPEI is polyplexed with the negatively charged pDNA, and the polyplex acquires a net positive charge, which makes it possible to introduce the polyplex into the cell. The lPEI is inexpensive and easy to handle, but also has a cytotoxic effect [15,16], necessitating careful application. Transient transfection of mammalian cells with lPEI is a conventional procedure in VLP production [17,18,19,20]. However, to the best of our understanding, the literature has not described a process for producing YF-VLP via lPEI-mediated transient transfection in HEK cells within a stirred-tank bioreactor. While some studies detail flavivirus VLP production using stable cell lines [21,22], the creation of such lines is not only time-intensive but also necessitates specialized equipment and incurs high costs. This makes it an impractical method for immediate responses to global outbreaks. Contrarily, transient transfection is cost-effective, rapid, straightforward, and does not demand any specialized apparatus.

The production of VLPs for a passive immunization is subject to the guidelines of good manufacturing practices (GMPs) and Process Analytical Technology (PAT). Such a YF-VLP production platform requires stable and robust processes that guarantee smooth production. Modern biotechnological processes rely on Process Analytical Technology to understand the process outcomes and, finally, to have data in order to control and automate the processes. PAT can be realized, e.g., by dielectric spectroscopy (DS). For DS, living biomass or cells are exposed to an alternating electric field, where they exhibit certain electrical properties, called permittivity (*ε*, in F/cm) and conductivity (σ, in S/m) at frequencies between 100 kHz and 10 MHz [23,24,25]. Both factors can be applied to living biomass, as permittivity describes the ability to store electrical energy, while conductivity dissipates it [24,25]. Since the cell membrane is non-conductive, the electrical charge (polarization) can build up inside the cell. The charge in the cytoplasm is built up by the dissolved salt ions and organic materials. The polarization of the cells can be represented as a function of the applied frequency. At a low frequency, the maximum polarization is present in the plateau. As the frequency increases, the polarization of the cell decreases to a lower plateau. This frequency-dependent phenomenon and the loss of cellular polarization is called β-dispersion, and its characteristic parameters are Δ*ε*, *f_C_*, and α [26]. The permittivity change Δ*ε* for spherical cells can be calculated according to Schwan 1957 [27] as follows:(1)$$\Delta \varepsilon =\frac{9\times V_{C\;}\times r\times C_{m}}{4}$$
where Δ*ε* is the permittivity [F/m], *r* is the cell radius [m], *V_C_* is the volume fraction of cells in a medium (%), and *C_m_* is the capacitance per membrane area (F/m^2^). The volume fraction of cells in a medium can be described as: (2)$$V_{c}=\frac{4\times \pi \times r^{3}\times N}{3}$$
where *N* is the cell density (cells/m^3^). The characteristic frequency *f_C_* (Hz), at which the polarization is half complete, depends on the cell size, the capacitance of the cell membrane *C_m_*, and the conductivity of the surrounding medium *σ_m_* (mS/cm), and cell cytoplasm *σ_c_* (mS/cm).
(3)$$f_{C}=\frac{1}{2\times \pi \times r\times C_{m}\times \left(\frac{1}{\sigma_{c}}+\frac{1}{2\times \sigma_{m}}\right)}$$

The empirical parameter α assumes values between 0 and 1, and reflects the heterogeneity of dielectric properties within the suspension culture. Graphically described, α reflects the slope of permittivity as the frequency is increased [28].

Therefore, DS is a suitable method to monitor and control the viable biomass in a production process [29,30,31,32,33]. Additionally, as the DS also gives information about the cell size/volume, the capacitance per membrane area (in form of Δ*ε* and *f_C_*), and the heterogeneity of the culture (in form of *α*), more information about transient VLP production processes can be obtained. DS in transient transfection enables an automated transfection based on the permittivity signal.

The degree of success of transfection depends on the cell cycle of the cell [18]. As demonstrated by Brunner et al., 2000 [34], transfection is more successful when cells are situated in the G2/M phase during the transient transfection process. During the G2 phase, HEK cells gear up for mitosis, undergoing swelling due to fluid absorption. This swelling can be indicated by a declining *f_C_* value since it’s inversely related to cell size. Subsequently, the M phase marks the cell’s division into two daughter cells, accompanied by the breakdown of the nuclear membrane. This dissolution of the nuclear barrier might enhance the introduction of foreign material during the M phase, offering a logical explanation for the increased transfection success observed during the G2/M phases. The understanding of this cell cycle-dependent transfection efficiency may be leveraged to optimize transient transfection processes for various applications, including the production of VLPs.

It may be possible to determine the time of the ideal transfection and to control and monitor the viable cell concentration. For lPEI, the N/P ratio can be varied, or the transfection reagent can be added over a longer period of time in fed-batch or continuous processes, to increase the productivity of the process under a full control of the process. Furthermore, this allows conclusions about the cell concentration, and possibly about the cell size and morphology. In addition to biomass, in production processes where cytotoxic substances such as lPEI may be used, the physiological state of the cell is also of interest for dosing lPEI, or for defining the correct harvesting time. DS has already been applied to various mammalian and insect cells, such as CHO, Vero, hybridoma, HeLa, Sf9, S2, and High five cells, to monitor the biomass and cell physiology during production processes [35,36,37,38].

The aim of our study was to develop a YF-VLP production process based on transient transfection in a stirred-tank bioreactor. Therefore, we first optimized the plasmid DNA (pDNA) production process, which is a prerequisite for the final YF-VLP production. Using a Plackett–Burman (PB) design, we screened different medium components, and identified the components that increase the specific pDNA yield and *E. coli* biomass. With an optimized medium composition, we were able to produce enough transfection-grade pDNA in shaking flasks. To set up the transient production process in the stirred-tank bioreactor, we carried out preliminary investigations. We used a HEK293T suspension cell line, lPEI as transfection agent, and a reporter plasmid to encode eGFP and YF-protein E in a serum-free medium, as described in [39]. Due to process efficiency, we investigated the minimally needed polyplexing time, cell density, and the need for a medium exchange after transient transfection that still reaches a high transfection efficiency. We further screened the effect of additives often used in STR set-ups, e.g., Pluronic F-68 and an anti-clumping agent, for transfection efficiency. The findings from these preliminary experiments were used to design an lPEI-mediated transient transfection in the STR in a batch mode. We further implemented the dielectric spectroscopy as an inline PAT tool to monitor the process. We established a correlation between the viable cell concentration and permittivity. We determined the time for an ideal transfection, and identified relevant process events during cultivation, such as the uptake of the polyplex, a possible cell size change, nutrition starvation, the ideal harvest window, and the initiation of cell death and apoptosis. With this STR process, we then successfully produced and verified the YF-VLPs via Western blot, IFCM, and TEM.

## 2. Materials and Methods

### 2.1. Plasmid Production

The model plasmid pcDNA 3.1(+)-JEV-E-EGFP, used to optimize transient transfection, was 7.7 kb in size, and carried an ampicillin resistance for selection in bacteria. It also carried a reporter gene called enhanced green fluorescent protein (eGFP) to detect a successful transient transfection of HEK cells. Other properties of the plasmid are described by Dekevic et al. [39]. An exchange of the eGFP sequence, with the sequence for the protein prM, led to the plasmid pcDNA 3.1(+)-JEV-prM-E, and had the size of 7.0 kb. After establishing and optimizing the transient transfection with the model plasmid, using the reporter gene eGFP, the pcDNA 3.1(+)-JEV-prM-E plasmid was used to produce YF-VLPs.

#### 2.1.1. pDNA Production under Standard Conditions

NEB 10-β competent *E. coli* were thawed on ice, and 100 ng of pDNA were added to 50 µL of the thawed *E. coli* cells. The mixture was incubated on ice for 30 min and heat-shocked at 42 °C for 30 sec to ensure the uptake of the pDNA into the cells. The tube was placed on ice for 5 min before adding 950 µL LB (*Luria-Bertani*)-medium (Carl Roth GmbH & Co. KG, Karlsruhe, Germany) to the mixture. The cells were then incubated at 37 °C for 60 min and shaken vigorously at 250 rpm in a thermomixer (Eppendorf, Hamburg, Germany). A total of 100 µL of transformants were then spread on selection plates (LB-agar plates) with 50–100 µg/mL ampicillin (Carl Roth^®^ GmbH & Co. KG, Karlsruhe, Germany) and incubated overnight at 37 °C. Single clones were picked the next day and cultured in a 3 mL LB-medium with 50–100 µg/mL ampicillin. The following day, the transformed *E. coli* were further cultured in baffled 1000 mL shake flasks with a 200 mL working volume at 250 rpm and 37 °C.

#### 2.1.2. Optimization of pDNA Production Using a Plackett–Burman Design

We screened individual media components, using a Plackett–Burman design with 12 ((*n*_1_ + *n*_2_) + 1) runs, 9 (*n*_1_) factors, and 2 (*n*_2_) dummies, to estimate the occurring falsification related to cumulative interaction components, as shown in Table 1. We optimized the process with regard to a maximum specific pDNA yield (ng pDNA/ODV) (OD_600_ × culture volume = ODV) and the *E. coli* biomass (determined as OD_600_). The following components were used for this purpose: LB powder (consists of 10 g/L trypton, 5 g/L yeast extract, 10 g/L NaCl) (Carl Roth), meat peptone (Fluka Analytical), casein peptone (Carl Roth), yeast extract (Carl Roth), glucose (Carl Roth), glycerol (Carl Roth), phosphate (Carl Roth), NaCl (Carl Roth), and MgSO_4_ (Carl Roth). Experiments were performed in 500 mL shake flasks and harvested and tested after 18 h.

#### 2.1.3. pDNA Purification

To purify the pDNA and to obtain transfection-grade pDNA, we used the NucleoBond PC-10000 plasmid purification protocol (Macherey-Nagel GmbH). In short, we centrifuged (Sigma, Osterode, Germany) (Rotor: 12024-H/16,000× *g*) the bacteria culture in an ODV, as defined by the manufacturer, with 6000× *g* for 15 min at 4 °C. The cell pellet was lysed, and the pDNA was finally precipitated using isopropanol. The pDNA pellet was dried under sterile conditions, and dissolved in sterile deionized H_2_O.

### 2.2. eGFP or YF-VLP Production

#### 2.2.1. Cell Line and Medium

We cultivated the suspension cell line HEK 293T/17 SF (#ATCC ACS-4500) in a serum-free FreeStyle 293 medium (Thermo Fisher Scientific, Waltham, MA, USA) with 10 mL/L Insulin-Transferrin-Selenium (ITS) in 100-mL baffled shake flasks with a 20-mL working volume. An animal origin-free and chemically defined anti-clumping agent, ACA (Thermo Fisher Scientific), was first added in a ratio of 1:800 to reduce cell aggregation and to reach higher viable cell densities during passages, and then removed at least 24 h prior transient transfection. The baffled shake flasks were incubated at 37 °C in a humidified 8% CO_2_ atmosphere, shaking at 100 rpm. Routinely, passages were performed at a concentration of 0.3–0.5 × 10^6^ cells/mL.

#### 2.2.2. Transient Transfection in Shaking Flasks

To transfect the cells during the exponential growth phase, they were resuspended in FreeStyle 293 medium plus ITS at a concentration of 1 × 10^6^ cells/mL for at least 24 h before transfection. For this purpose, the cells were pelleted by centrifugation (300× *g*, 5 min, room temperature) and then resuspended in a fresh medium. We transiently transfected the cells with 1.1 pg pDNA/cell with the plasmid pcDNA 3.1(+)-JEV-E-EGFP and 6.6 pg lPEI/cell [39]. For transfection, we used a 1 mg/mL 25-kDa lPEI solution (Polyscience Europe), pH 5. The transfection volume was 10% of the total sample volume, and was prepared by adding the appropriate amount of lPEI to the pDNA (both dissolved in a freshly supplemented room-temperature medium). The polyplexes were formed for different time intervals (between 5 and 20 min). The transfected cultures were incubated at 37 °C in a humidified 8% CO_2_ atmosphere, shaking at 100 rpm, in 100 mL baffled shake flasks with a working volume of 20 mL.

#### 2.2.3. Transient Transfection in a Stirred-Tank Bioreactor

We used a 2-L B-DCU bioreactor (Sartorius, Göttingen, Germany) with a working volume of 1 L. We used a double jacket glass vessel, equipped with a three-blased pitched-blade impeller, with a diameter of 54 mm and an angle of 45°. The agitation was set to 100 rpm. The temperature was maintained at 37 °C with a PT100 probe (Sartorius, Goettingen, Germany), and the pH was maintained at 7.2 (Hamilton, Bonaduz, Switzerland) with 1 M NaOH and CO_2_ pulsation. The dissolved oxygen (DO) (Hamilton, Bonaduz, Switzerland) was maintained at 50% air saturation, by sparging with 100% O_2_ through a common ring sparger. An Incyte DS probe (Hamilton, Bonaduz, Switzerland) was used to measure the viable cell concentration. The DS probe recorded the data every 2 min in a range between 300 kHz and 1 MHz.

Before autoclaving, the pH probe was calibrated. The FreeStyle 293 medium with ITS/F-68 supplements was added to the sterilized bioreactor setup in a volume of 800 mL, and heated to 37 °C overnight for sterility testing. DO and DS probes were calibrated in a fresh, supplemented, and heated medium before inoculation. The reactor was inoculated with 100 mL fresh, supplemented, and heated medium, with cells in a mid-exponential growth phase, at a starting concentration of 0.5 × 10^6^ cells/mL, and a viability >90%. Pluronic F-68 (Thermo Fisher Scientific) non-ionic surfactant was added, in order to culture in a concentration as stated at the experiment, to control the influence of shear forces in suspension cultures, to reduce foaming, and to reduce the cell attachment to bubbles and to the glass. The transient transfection was performed at a cell concentration of 1 × 10^6^ cells/mL. We transiently transfected the cells with 1.1 pg pDNA/cell with the plasmid pcDNA 3.1(+)-JEV-E-EGFP or pcDNA 3.1(+)-JEV-prM-E, and 6.6 pg lPEI/cell. For transfection, we used a 1 mg/mL 25-kDa lPEI solution, pH 5. The transfection volume was 10% of the total reactor volume, and was prepared by adding the appropriate amount of lPEI to the pDNA (both dissolved in a freshly supplemented room-temperature medium). The polyplexes were formed for 5 min at RT.

### 2.3. Analysis

#### 2.3.1. pDNA Analysis

The pDNA concentration was measured using the Quant-iT^TM^ PicoGreen assay (Invitrogen Quant-iT^TM^ PicoGreen^TM^ dsDNA Assay Kit, Thermo Fisher Scientific, Karlsruhe, Germany). The purity was determined by measuring the absorption at A_260/280_ with an UV-spectroscopy (Cytation™ Cell Imaging Multi-Mode Reader + CO_2_/O_2_ Gas Controller, BioTek™ Instruments, Inc., Bad Friedrichshall, Germany).

For restriction analysis, 300 ng of pure plasmid were incubated for at least 30 min at 37 °C with none, one (PflMI), or two (PflMI and BglII) restriction enzyme/s (0.2 µL) and the NEBuffer™ 3.1 (2 µL) (New England Biolabs, Frankfurt am Main, Germany). The samples were separated in a 1% agarose gel with SYBR safe at 130 V, 2 A for 35 min in a horizontal chamber in a TAE buffer. Subsequently, the gel was analyzed via ChemiDoc™ version 1.0 and ImageLab^TM^ version 6.0.1 software.

#### 2.3.2. Analysis of *E. coli* Growth

We measured the OD_600_ with a photometer (BioSpectrometer kinetic, Eppendorf AG, Wesseling-Berzdorf, Germany) according to the manufacturer’s instructions. For a further calculation of the specific pDNA yield, we normalized the OD_600_ to the volume, obtaining the ODV = OD_600_ × Volume. 

For pH measurements, the samples were centrifuged for 3 min at 4 °C, and with 16,100× *g* of the supernatant were measured using the pH-meter (FiveEasy, Mettler-Toledo, Giessen, Germany).

#### 2.3.3. Offline Determination of Cell Concentration

Fluorescence-activated cell sorting (FACS) in a Guava easyCyte benchtop flow cytometer (Luminex, Austin, TX, USA) was applied to determine cell concentration and viability, using propidium iodide according to the manufacturer´s instructions.

#### 2.3.4. Metabolite Analysis

The ammonium concentration was measured with the Cedex Bio Analyzer (Roche AG) from the supernatant of the *E. coli* culture, according to the manufacturer´s instructions, after centrifugating the sample for 3 min, 4 °C and with 16,100× *g*.

The HEK cells were centrifuged for 5 min with 400× *g*, and glucose and lactate measurements were analyzed from the supernatant, using a Biosen C enzymatic-amperometric analyzer (EKF Diagnostic, Barleben, Germany) according to the manufacturer´s instructions.

#### 2.3.5. SDS-PAGE

For SDS-PAGE, a 4–10% Criterion^TM^ TGX Stain-Free^TM^ gradient gel (Bio-Rad, Feldkirchen, Germany) with Tris-Glycine-SDS running buffer (25 mM Tris, 192 mM Glycine, 0.1% SDS, pH 8.3) was used. The samples were diluted 1:4 with the loading buffer (4× Laemmli Sample Buffer containing 10% volume β-mercaptoethanol, Bio-Rad) and denatured at 95 °C for 5 min. The Precision Plus Protein^TM^ Unstained Standard (10–250 kDa) and Precision Plus Protein^TM^ WesternC Standard (10–250 kDa) (Bio-Rad, Feldkirchen, Germany) were used as standards. Electrophoresis ran for 25 min and at 250 V. The protein bands were visualized in ChemiDoc^TM^, Bio-Rad. 

#### 2.3.6. Western Blot

For WB, the protein transfer from SGS-gel to the ready-to-use polyvinylidene difluoride membrane (Trans-Blot Turbo^®^ Midi PVDF Transfer pack, Bio-Rad) was performed for 7 min, 2.5 A and 25 V, according to the manufacturer’s instructions, using the Trans-Blot^®^ Turbo^TM^ Transfer System (Bio-Rad). The subsequent blotting of the membrane was performed with a 5% BSA solution (Carl Roth) under continuous swirling for 1 h at RT, and finally washed three times in PRBS containing 0.1% Tween 20^®^ (PBS-T) (Carl Roth, Karlsruhe, Germany). The membrane was probed with a primary antibody (rabbit, 1:1000) in PBS, containing 0.05% Tween 20, at 4 °C overnight, with continuous panning. The following day, the membrane was washed three times and incubated with the secondary antibody (Goat-Anti-rabbit, 1:5000, Thermo Fisher) and StrepTacin-HRP (1:10,000, Bio-Rad) for 3 h at RT. The membrane was washed three times and prepared for analysis by adding equal amounts of both solutions of Clarity Western ECL Substrate (Bio-Rad) to the membrane, incubating for 5 min in the dark. The visualization of the bands was realized in ChemiDoc^TM^, Bio-Rad.

#### 2.3.7. Imaging Flow Cytometry Measurement (IFCM)

90 µL of FreeStyle medium, the VLP-containing supernatant of the bioreactor (P01), and the clarified supernatant, either centrifugated at 400× *g* for 10 min (P02), or centrifugated at 10,000× *g* for 15 min (P03), were incubated without any further processing, with 0.1 µL of Alexa Fluor 647 conjugated Yellow Fever virus prM Protein antibody (IgG GTX133957) in 10 µL PBS for 2 h at room temperature. According to the MiFlowCyt-EV criteria [40], the buffer only, buffer/medium plus antibody control, and the fluorochrome-conjugated isotype antibodies were used as controls. All samples were analyzed in technical duplicates, using the built-in autosampler from U-bottom 96-well plates (Corning Falcon, cat 353077) with a 5 min acquisition time per well on the AMNIS Image Stream X Mark II Flow Cytometer (AMNIS/Luminex, Seattle, WA, USA). All data were acquired at 60× magnification at a low flow rate (0.3795 ± 0.0003 μL/min, directly determined by the system) and with the removed beads option deactivated, as described previously [41,42]. The data were analyzed as described previously [43]. 

#### 2.3.8. Transmission Electron Microscopy (TEM)

TEM analyses were performed exactly as described recently [44]. Briefly, 7.5 µL of 1:100 diluted VLP preparations were mixed with 1 µL aqueous contrasting solution, containing 1% methyl cellulose (*w/v*; Sigma Aldrich) and 2% uranyl acetate (*w/v*; Polysciences, Warrington, FL, USA). Following incubation for 10 min, 0.5 µL droplets were placed on 200 mesh copper grids (Plano, Wetzlar, Germany) and dried at room temperature, allowing included VLPs to adhere to the films’ surfaces. Images were taken on a JEM 1400 Plus electron microscope (JEOL, Tokyo, Japan), equipped with a 4096 × 4096 pixel CMOS camera (TemCam-F416; TVIPS, Gauting, Germany), and run at an operating voltage of 120 kV. Image acquisition software EMMENU (Version 4.09.83) was used for taking 16-bit images. Image post-processing was performed with the software ImageJ (Version 1.52b; National Institutes of Health, Bethesda, MD, USA).

#### 2.3.9. Statistical Analysis

A two-way analysis of variance (ANOVA) was performed to determine the statistical significance among different parameters using GraphPad Prism Version 5.01 software. Statistical differences were defined as follows: * *p* < 0.05, ** *p* < 0.01, *** *p* < 0.001. The experiments were performed for at least three independent determinations, and presented as the mean value ± standard deviation (mean ± SD), unless stated otherwise.

## 3. Results

In our endeavor to optimize the production of YF-VLPs using a bioreactor-centric approach, we meticulously evaluated various parameters. These included the conditions for plasmid DNA production, the transient VLP production dynamics within the stirred-tank bioreactor, and the implications of potential additives. Here, we present our key findings, detailing the significant impacts on the transfection efficiency and the subsequent validation of the YF-VLPs produced.

### 3.1. Optimization of Transfection-Grade pDNA Production

For a transient transfection on a bioreactor scale, pDNA concentrations above 6.6 mg pDNA/L are needed. We first optimized the pDNA production in *E. coli* by optimizing the production medium. Our reference was the *E. coli* biomass (in OD_600_) and the pDNA yield obtained in a LB medium (OD_600_ 2–3, pDNA < 1 mg/L). We screened the following medium components, which should increase *E. coli* biomass and/or specific pDNA yield, using a Placket–Burman experimental design: LB powder, meat peptone, casein peptone, yeast extract, glucose, glycerol, Na_2_HPO_4_/KH_2_PO_4_ phosphate buffer, NaCl, and MgSO_4_. Although less than 0.6% of the possible combinations are investigated, the PB design is extremely efficient, as the significant factors are usually found [45]. In our screening, we determined that NaCl and phosphate increased the OD_600_ and the specific pDNA yield. The addition of glucose decreased the OD_600_, but interestingly increased the specific pDNA yield. The *E. coli* biomass was further increased by yeast extract, with no impact on the specific pDNA yield (Figure 1).

Therefore, we used this optimized media composition (10 g/L glucose, 10 g/L NaCl, 10 g/L yeast extract, either with 75 mM or 150 mM Na_2_HPO_4_/KH_2_PO_4_ buffer) and determined the pDNA production kinetics to define the optimal time of harvest. The concentration of the Na_2_HPO_4_/KH_2_PO_4_ buffer had a significant impact on the specific pDNA production. In the 75 mM phosphate-buffered culture, the specific pDNA production was very low (48.62 ± 15.93 ng pDNA/ODV, 27 h), whereas the specific pDNA production in the 150 mM phosphate-buffered culture reached a maximum of 947.382 ± 163.377 ng pDNA/ODV (n = 3) after 27 h. The optimal harvest time window was between 24 and 27 h. Using the optimized production medium and the optimal time of harvest, the pDNA concentration increased more than eleven-fold to 11 mg/L.

Interestingly, the *E. coli* growth kinetics were similar under both conditions, with a maximum OD_600_ of 13.2–13.8 after approximately 7 h (Figure 2A), and growth rates of µ_max_ = 1.2022 ± 0.0111 1/h (75 mM phosphate buffered culture) and 1.1851 ± 0.0051 1/h (150 mM phosphate buffered culture). Also, the glucose consumption was only slightly different for both media, as glucose was completely consumed after 9–11 h (75 mM phosphate-buffered culture), or 6–7 h (150 mM phosphate buffered culture). For both media, we observed an initial increase in the NH_3_ concentration to 4–5 mmol/L, followed by a decrease to 0.5 mmol/L until reaching the stationary growth phase. In the stationary growth phase, the NH_3_ concentration increased again, reaching final values of 1.798 ± 0.213 mmol/L for the 75 mM buffered system, and 2.83 ± 0.16 mmol/L for the 150 mM buffered system after 27 h. The largest difference (besides the pDNA yield) was found in the course of the pH value in both media. The pH value in the 75 mM phosphate buffered culture dropped fast to pH 5.1 between 3 and 10 h cultivation time, and stayed constantly low. The pH value of the 150 mM phosphate-buffered culture decreased more slowly, reaching pH 6.0 after 8 h, and then increased again, until the final pH value of 6.9 was reached after 27 h.

As it is known that temperature may have an impact on *E. coli* growth and pDNA production [46], we investigated the influence of temperature (33 °C, 35 °C, 37 °C, and 39 °C), expecting that a reduced growth rate at lower temperatures would lead to higher pDNA yields. We found that the specific pDNA production deteriorated significantly at temperatures lower than 37 °C, reaching only <200 ng pDNA/ODV. On the other hand, a cultivation at 39 °C resulted in a slightly higher OD_600_, but a lower and less robust specific pDNA yield, compared to a production at 37 °C. For this reason, we continued the pDNA production at 37 °C (Figure A1 in the Appendix A).

With the optimized set-up, we produced pDNA and purified it. We found that the pDNA was pure, intact, and of the correct size (Figure A2 in the Appendix A). In addition, we confirmed the correctness of the pDNA sequence using Sanger sequencing.

In our shake flask experiments, we were able to produce a yield of 11 mg/L of pure transfection-grade pDNA. For transient transfection in the bioreactor, the required 6.6 mg pDNA/L could thus be covered with a total volume of 600 mL *E. coli* culture.

### 3.2. Optimization of Process Parameters for a Transient VLP Production in a Stirred-Tank Bioreactor

To establish an efficient transient transfection process on a bioreactor scale, we carried out small-scale experiments to identify relevant process parameters that influence the transfection efficiency. Our aim was to reach a high transfection efficiency, while minimizing cell concentration, time, and additional process steps. We first investigated the influence of different cell densities, and found that higher transfection efficiencies were achieved with lower cell densities, compared to our standard procedure (2 × 10^6^ cells/mL) (Figure 3A). We further reduced the polyplex formation time (Figure 3B) from 20 min to 10 min or 5 min, and found no significant changes in the transfection efficiency. We then investigated whether a medium exchange was necessary to minimize the influence of excess and cytotoxic lPEI on cells. Therefore, due to the duration of the transfection process of 1.5–4 h, we analyzed the effect of a medium exchange at 2 hpt, 4 hpt, and 6 hpt (hours post transfection), and compared it to a culture that was transfected without a subsequent medium exchange (Figure 3C). Interestingly, the medium exchange did not generate a significant improvement in the transfection efficiency. 

We further investigated the effect of additives. We investigated the effect of Pluronic F-68, which is often used in STR processes to protect mammalian cells against shear forces. Furthermore, the HEK cells tend to form aggregates in suspension cultures, which can be prevented by adding an anti-clumping agent. However, both additives might influence the transfection, as they alter the cell membrane properties, and may also counteract with the polyplex. We found that F68 concentrations of 0.025% and 0.05% (*w/v*) had no significant effect on the transfection efficiency. In contrast, we found a strong impact of even low concentrations of ACA on the transfection efficiency (Figure 3E). Even the smallest amounts of ACA led to a total loss in transfection efficiency. To avoid a strong agglomeration, we investigated the addition of ACA 4 hpt, and again measured the same high transfection efficiencies (Figure 3D). For further experiments, the addition of ACA before a transient transfection was excluded. If ACA was added, it was added either 4 hpt in shake flasks, or 24 hpt in the bioreactor. We also analyzed the effect of additives on HEK cell growth, finding no differences (reference µ = 0.0272 ± 0.0006 1/h, addition of 0.025% F68—µ = 0.0265 ± 0.0014 1/h, addition of 2 mg/L ACA—µ = 0.02619 ± 0.00007 1/h). Only the addition of 0.05% F68 led to a significantly lower growth rate of µ = 0.0237 ± 0.0012 1/h, and to a highly significant increase in the lactate formation rate.

Therefore, the STR process was set up using the following parameters: a polyplex formation time of 5 min, a transient transfection at a cell concentration of 1 × 10^6^ cells/mL, no medium exchange, an addition of 0.025% F68, and an avoidance of ACA (or an addition 24 hpt).

### 3.3. Monitoring a Transient VLP Production in a Stirred-Tank Bioreactor Using Dielectric Spectroscopy

We transferred the transient transfection process to the STR. Standard parameters such as pH, temperature, and dissolved oxygen were kept constant, and the transient transfection was carried out with the reporter plasmid pcDNA 3.1(+)-JEV-E-EGFP. The HEK cells were first grown to a cell concentration of 1 × 10^6^ cells/mL, and then transfected using 1.1 pg pDNA/cell and 6.6 pg lPEI/cell, based on our optimizations published before [39]. Literature data [47,48,49] supposed the highest product titers 24–96 h after transient transfection. First, we measured eGFP-positive cells 24 hpt, using the plasmid to encode the reporter protein eGFP. eGFP was expressed within the cells (Figure A7 in the Appendix A), reaching a maximum transfection efficiency of 41% 48 hpt (Figure 4A1).

We used inline dielectric spectroscopy (DS) to monitor the process. DS is a powerful technique that measures the properties of the cells, and can be correlated to the viable biomass [23,37,38]. We correlated the offline cell concentration data in the exponential growth phase with the measured permittivity *ε* of five runs to determine inline, at which time the optimal cell concentration of 1 × 10^6^ cells/mL for transfection (Figure A6 in the Appendix A) is reached. The coefficient of determination was 86%. The correlation of the cell concentration measured offline and inline was significantly more accurate before transient transfection than after. As soon as the culture entered the stationary growth phase, the in- and offline cell concentration measurements became more inaccurate.

DS is also able to detect morphological changes in the cells. It was possible to monitor the attachment of the polyplex to the cell membrane with DS. After the addition of the polyplex solution, the permittivity first dropped (due to dilution), and then further decreased slowly over 2–2.5 h (Figure 4A1). This correlates with the membrane attachment and uptake time of the polyplex towards the HEK cells [39], and could reflect the changes in the cell membrane, which occur during the polyplex uptake. A few hours later, the permittivity signal increased steeply again, reaching a constant value at the end of the transfection process.

The β-dispersion is a frequency-dependent phenomenon, whose characteristic parameters are *f_C_*, Δ*ε*, and α, which allow cell physiological interpretations of the culture. 

After transfection, the *f_C_*-profile also increased for 2–2.5 h, which may also reflect the uptake of the polyplex. The *f_C_* signal then decreased again after 120 h. At this time point, (i) glucose was completely depleted, and (ii) the transfection efficiency was at a maximum. Subsequently, the signal increased until the end of the cultivation, maybe reflecting changes in cell size/membrane polarization due to cellular death. The Δ*ε*-curve also showed a characteristic increase, with a short dip at the time of transient transfection (Figure 4A3). The maximum value for Δ*ε* was reached shortly after the glucose supply was depleted and the transfection efficiency was at a maximum. Subsequently, the signal dropped again. The course of the α-curve (Figure 4A3) was characterized by a slight and steady increase after about 120 h. Subsequently, the signal entered a plateau, and remained there until the end of the cultivation.

We applied the same process of STR set-up for the VLP production, except that we used the YF-VLP plasmid pcDNA 3.1(+)-JEV-prM-E. With the addition of ACA after 94 h (Figure 4A), the culture, transfected with the reporter plasmid pcDNA 3.1(+)-JEV-E-EGFP, reached a higher maximum cell concentration of over 3 × 10^6^ cells/mL, compared to the culture with the YF-VLP plasmid (under 2 × 10^6^ cells/mL). The glucose consumption was in accordance with the growth profile. The culture transfected with the reporter plasmid pcDNA 3.1(+)-JEV-E-EGFP, which reached higher cell density and depleted glucose after 90 h to 110 h (Figure 4A2), whereas the other culture consumed the glucose completely after 120 h (Figure 4B2). With a complete glucose depletion, the cultures entered the stationary growth phase. Lactate was produced proportionally to the glucose consumption. The culture transfected with the reporter plasmid pcDNA 3.1(+)-JEV-E-EGFP started to consume lactate after the glucose depletion.

Using two different plasmids (reporter plasmid pcDNA 3.1(+)-JEV-E-EGFP (Figure 4A1) and YF-VLP plasmid pcDNA 3.1(+)-JEV-prM-E (Figure 4B1), the course of the permittivity changed, especially at the end of transfection. The first 3 h after transfection were very comparable, then differences occurred in permittivity, Δ*ε*, *f_C_* and *α* signal. During the YF-VLP production, we did not detect the clear maximum in permittivity and Δ*ε*, which may reflect the changes occurring due to the release of the VLPs, which are completely different from the intracellular GFP production. The *f_C_* signal was more-or-less constant throughout the VLP production, and the α-curve of the VLP production showed a continuous decrease, only interrupted by a short increase around 96 h.

However, plotting the Δ*ε* against *ε* showed a very similar picture for both processes (Figure 5). The course of the Δ*ε*-*ε*-plots initially showed an almost exclusive linear relationship between Δ*ε* and *ε* in the exponential growth phase. At the time of transient transfection, there was an approximately 10% shift in the plot at the *ε* level, due to the dilution of the culture broth with the addition of the transfection reagent. For another 2.0–2.3 h, there was a drop in the Δ*ε*-signal until the signal started to increase again, with a parallel shift on the *ε* level. This parallel shift was recorded in all our cultivations with a lPEI-mediated transient transfection. The further course after transient transfection was largely linear. Different events in the process, such as a complete glucose consumption, were characterized by a change in the slope, and a higher signal noise. The end of the exponential growth phase was also particularly striking. Therefore, there was a characteristic turn in the signal to lower Δ*ε* values, which aligned with the optimal time to harvest.

Using the permittivity signal, we calculated the growth rates over the entire cultivation period in different intervals of 5 h, 10 h, and 24 h, as shown in Figure A8 in the Appendix A. It turned out that the higher resolution of the intervals of 5 h or 10 h, compared to 24 h, led to a significant gain in information on the transfection process and its connection with the growth rate. For example, it was characteristic of the transient transfection with the cationic lPEI that the growth rate dropped significantly into the negative range of up to −0.6 1/h immediately after the transfection. This fall in growth rate was particularly evident at the higher resolution of the 5 h and 10 h intervals, and this effect occurred consistently. With the calculated growth rate in the interval of 24 h, or in the offline data, the extent was not visible, or only with a considerable delay.

### 3.4. Characterization of Produced YF-VLPs

We analyzed the cell culture supernatant every 24 h after transfection, with a Western blot to detect the presence of the two YF structural proteins, E and M. We could not clearly determine the presence of protein E, as all antibodies we tested bound nonspecifically, resulting in several bands, or even bands in the negative control. In contrast, protein M (and its preform preM) was clearly detected after 26 hpt. The concentration of protein M increased and reached its maximum at 96 hpt, when the STR was harvested (Figure A10 in the Appendix A). The non-transfected HEK 293T/17 SF cells showed no specific protein M expression (Figure 6).

For the model system, we used the reporter gene eGFP to optimize transient transfection. To quantify the YF-VLP produced, we used a method called imaging flow cytometry measurement (IFCM) (Figure 7), which is not commonly used for VLP detection but is still very specific. This method, which is primarily used to quantify extracellular vesicles [43,50], allowed very specific detection of one of the surface proteins named prM/M of the YF-VLPs. The YF-VLPs produced were labeled with a fluorochrome-conjugated prM antibody, which made it possible to determine a particle concentration of 2.23 × 10^7^ ± 0.43 × 10^7^ labelled particles/mL in the reactor supernatant. After centrifugation at 10,000× *g*, 8.375 × 10^6^ ± 2.625 × 10^6^ labelled particles/mL remained in the culture supernatant. TEM is a standard method for characterizing VLPs and was used to determine the size of the YF-VLPs. The evaluation of the TEM image in Figure 7D shows VLP structures with an average size of 66.0 ± 17.4 nm (n = 93).

## 4. Discussion

The success of producing YF-VLPs through a bioreactor-centric approach hinged on the careful optimization of various parameters. A crucial element in this process was the production of plasmid DNA (pDNA), given its role in transient transfection. This study delved into the detailed optimization process of pDNA production, which included the use of statistical tools like the Plackett–Burman design of experiments (DoE), the assessment of essential medium components, and the careful balancing of conditions to maximize yield without undesirable byproducts. Further exploration of the influence of different process parameters on transient transfection highlighted the complexity of the process. It opened a path to understanding the delicate interplay between various factors, from polyplexation time to cell density effects. The findings and insights discussed here elucidated the methodology behind our successful YF-VLP production and provided valuable guidance for future applications in passive immunization techniques.

### 4.1. Impact of Various Parameters on pDNA Production in E. coli

We had to optimize the pDNA production, as we needed more than 6 L of *E. coli* culture to produce the 6.6 mg of pDNA required to transfect the cells in one STR run. With the Plackett–Burman design of experiments (DoE) approach, we quickly and effectively identified the medium components that increased the *E. coli* biomass and the specific pDNA yield. This confirms the power of DoE as a statistical tool to optimize processes [51,52]. To increase the *E. coli* biomass, the main elements that build up biomass must be applied, which are mainly carbon (50–53% of bacterial dry weight), nitrogen (12–15%), hydrogen (7%), and phosphorous (2–3%) [53]. Moreover, different ions, e.g., Na^+^, K^+^, Cl^−^, and Mg^2+^, are needed to maintain the metabolism and as cofactors for some enzymes [54]. We tested different C-sources (glucose, glycerol), N-sources (peptides and amino acids in the meat peptone, casein peptone, and yeast extract), one P-source (Na_2_HPO_4_/KH_2_PO_4_), and two salts (NaCl, MgSO_4_). The use of excessively high glucose concentrations carries the risk of metabolic overflow (“Crabtree effect”) and undesirable acetate production. Acetate is a byproduct of glycolysis during a fast aerobic growth, and inhibits bacterial growth [55,56]. We also detected such an inhibitory effect of glucose on the *E. coli* growth in our experimental design. The use of glycerol, on the other hand, leads to less acetate production. The inhibitory effect is lower, which allows the use of glycerol in higher concentrations in batch cultures. We detected a positive impact of an N-source, but only for yeast extract. The other two peptones we tested had no or even a negative impact on the *E. coli* biomass. A P-source and NaCl increased the *E. coli* biomass, but also the pDNA yield, which is not surprising, as P is a main component of DNA, and NaCl highly increases DNA stability. Interestingly, although glucose decreased the *E. coli* biomass, it significantly increased the specific pDNA yield. Besides the culture media composition [57,58], many enhancers for a pDNA production are described in the literature [59], e.g., temperature/temperature-sensitive origins (e.g., pUC, pMM1, pMM7) [60,61,62], a plasmid backbone optimization, nutrients, the reactor mode of mostly batch or fed-batch [60,63,64], a reduced growth rate which increases the plasmid copy number [65,66,67,68,69], dissolved oxygen [70], and the origin of replication and host strain [59,71].

Carrying out pDNA production kinetics with our optimized medium composition, we found that glucose depletion (=C-source deficiency) in combination with a 150 mM Na_2_HPO_4_/KH_2_PO_4_ buffer increased the plasmid production. When the glucose concentration dropped to zero, the *E. coli* started to metabolize proteins, and produced ammonium as a side product, which we also detected in our set-up. Using the 150 mM Na_2_HPO_4_/KH_2_PO_4_ buffer, the pH of the *E. coli* culture remained higher, and could even attain a pH of 7 after glucose depletion (mainly due to ammonium production). Benefits of higher pH values around 8 on the ratio of the pDNA yield to biomass (Y_pDNA/x_) have been published, whereas acidic pH values of the culture lead to low µ and Y_pDNA/x_ [57]. This negative effect of an acidic pH on the pDNA yield was also detected in our pDNA production kinetics, and may be the result of the reduced proton motive force (PMF) and, consequently, a low ATP production [72,73].

Using a relatively simple PB approach, we reached a high pDNA yield of 11 mg/L with an *E. coli* biomass of 11.6 OD_600_ (=950 ng pDNA/ODV) in the shake flask compared to, e.g., O’Mahony et al. [74], who achieved a yield of 7 mg/L plasmid, and an OD_600_ of 8.5. It is likely that the pDNA yield can be increased even more by plasmid optimization, induction, or other strategies, as stated above.

### 4.2. Influence of Different Process Parameters on Transient Transfection

To intensify the process in the STR, we carried out preliminary investigations to reduce time, material (here cells), or process steps (here medium exchanges). We investigated the influence of polyplexation time on transfection efficiency. According to Dekevic et al. [39], the size of the polyplex ranges between ~300 nm (5 min) and 550 nm (20 min), all resulting in high transfection efficiencies in our approach. This was similar to other working groups, which detected little or no influence on the transfection efficiency at particular N/P-ratios [75]. Other groups reported a significant influence of the polyplexation time, and defined the best polyplexation time to be three to five minutes [76]. However, transfection without a prior DNA-PEI complex formation was also successfully performed [77,78]. It is rarely possible to compare the studies in the literature, because they differ greatly in terms of the cultivation system used, the medium, and its composition, including osmotic pressure, the transfection reagent, the plasmid, and the temperature of the medium. We further investigated the impact of the cell density effect, which describes the decrease in cell-specific productivity/transfection efficiency as soon as a certain cell concentration is exceeded. The cell density effect mainly occurs at a higher cell density of about 1–4 × 10^6^ cells/mL. We observed this effect at densities ≥1.5 × 10^6^ cells/mL, despite a change of medium prior to transfection, to prevent interactions of cell metabolites with the transfection reagent. We further only used cells, which were in an early exponential growth phase at the time of transient transfection, to neglect any effect arising from the cell status. We could exclude that the cell density effect is caused by the presence of inhibitory components in the culture medium, preculture, and cell physiological conditions, as described by another group [79]. The discussion of the reasons for the cell density effect is still ongoing, and the exact mechanism of the effect still remains unclear [20,80,81].

HEK suspension cells tend to form aggregates, which can be increased by lPEI-mediated transient transfection [82]. The addition of the anti-clumping agent was primarily applied to prevent an aggregation of the HEK suspension culture and, thus, to achieve higher viable cell densities and viabilities. It is known that ACA has a significant negative influence on transient transfection processes with cationic polymers. The effect of ACA, to separate cells and avoid or minimize cell aggregation, might be due to changes in the surface charge of the cells. ACA may integrate into/adhere to the cell membrane, so that a positively charged polyplex has no possibility to attach to the membrane, or may even be actively repelled due to equal charges. We tested whether a very low concentration of ACA can still hinder agglomeration, but also allow sufficient transfection. However, even the lowest ACA working concentration led to a complete inhibition of a transient transfection in our set-up. Conceivably, ACA could be added several hours after transfection, to break up cell clumps and to maintain viability.

### 4.3. Implementation of Dielectric Spectroscopy into a Transient YF-VLP Production Process

Implementing dielectric spectroscopy in a transient production process, the simplest readout is the cell concentration, if a correlation was established. In our experiments, the viable cell concentration in the exponential phase could be well correlated to the permittivity signal, as long as exponential HEK cell growth occurred. The literature also showed that the correlation of living biomass to permittivity is significantly higher in the exponential phase than in the later cultivation phase [37,83,84]. A correlation of an offline-determined cell concentration with a permittivity signal is difficult, as the measurement principle of the inline and the offline method are fundamentally different. We correlated the permittivity signal with the offline cell concentration, determined with propidium iodide staining and flow cytometry. The calibration is therefore only valid for these two methods, and not for any other cell concentration method.

After the addition of the transfection polyplex, the permittivity signal was impacted by other phenomena. First, lPEI is cytotoxic and might change the membrane capacitance of the cell. Second, the adsorption of the polyplex, which often has a strong positive zeta potential of 12–35 mV [85,86,87], changes the net charge of the cell membrane and, as a consequence, affects the permittivity signal. In our experiments, we determined a decreasing permittivity signal and the time frame (1.84 h to 2.83 h) that exactly fits the uptake time of the polyplex. A similar drop in the permittivity signal immediately after an lPEI-mediated transient transfection was also seen by Ansorge et al. [88]. Third, cells tend to strongly agglomerate after transfection. This strongly influences the permittivity signal and makes reliable signal interpretation difficult [89]. Ron et al. [90] investigated the influence of cell agglomeration on the dielectric characteristics via theoretical models. In comparison to the homogenous suspended single spherical cells, the agglomerated cells showed a significant decrease in the permittivity signal and a shift in the critical frequency *f_C_*.

According to Equation (3), a change in the *f_C_* curve can be a consequence of a change in cell size, membrane capacitance *C_m_*, or in the conductivity of the medium or cytoplasm *σ_C_*. One explanation is that the decrease in the *f_C_* value after 24 hpt resulted from an enlargement of the cells. Another explanation is given by Ansorge et al. [91,92]. They showed that the *f_C_* value provides information about the availability of nutrient sources, which, in turn, has an influence on the intracellular conductivity *σ_C_*. Since glucose was depleted after 24–48 hpt in our experiment, and the *f_C_* signal then started to decrease, we could confirm this observation. The increase in the *f_C_* signal after 48 h was probably due to the fact that the HEK cells started to metabolize lactate, which is already known from the literature [93]. However, studies have also shown that a significant increase in *f_C_* levels towards the end of cultivation indicates cell death and apoptosis [94,95].

The α value of 0 represents a hypothetically perfect capacitor, whereas 1 for *α* represents a hypothetically perfect resistor. α reflects the heterogeneity of dielectric properties in the suspension culture, and increases as the distribution of the electrical properties of the cells expand. González-Correa et al. [96] showed that a higher α value is possibly related to a higher permeability of the membrane to ions.

Any change in the Δ*ε*-*ε* plot can be attributed to an important event in the cultivation. For example, a transfection reagent addition resulted in a direct *ε*-shift to smaller *ε*-values, before the Δ*ε*-*ε* plot turned linearly negative for 138 min. These 2 h approximately correspond to the time required for the polyplexes to penetrate the cell membrane and enter the cell interior. Glucose starvation represented another event in the cultivation, causing the Δ*ε*-*ε* signal to noise more strongly and assume a higher slope. A turn in the curve occurred at approximately 115 h in the model system, shortly after glucose was depleted. While the *ε*-value remained relatively constant, the Δ*ε*-value decreased from about 50 pF/cm to about 37 pF/cm by the end of cultivation. The characteristic *f_C_* value started to increase at approximately 120 h, and it could be assumed that apoptosis was initiated, which is probably shown by the inflection point in the Δ*ε*-*ε* plot. The Δ*ε*-*ε* plot for a VLP production shows much smaller Δ*ε*- and *ε*-values, as the cell concentration hardly increased after transient transfection, probably because the cells strongly aggregated and were not dissociated by an addition of ACA. However, the progression trend is very similar to the model system, except for the inflection point at the end of culturing, which was absent here. It can be assumed that this event has not yet occurred, because the VLPs were harvested after 123 h (96 hpt).

Dielectric spectroscopy allowed a calculation of growth rates at shorter intervals (5 h) than by the offline data (24 h). The decrease in growth rates to negative values immediately after transient transfection meant that the cell number decreased after transfection. This effect is probably due to the cytotoxicity of the lPEI, as it can lyse cells and release their cellular debris and sticky DNA, causing the cells to form large aggregates, and to decrease the single cell concentration in the culture significantly.

A major difference between the model system with eGFP expression and the YF-VLP production process is that the model system does not produce VLPs that are secreted from the cell. This is a significant step in the cultivation process. Furthermore, in the model cultivation, ACA was added after transient transfection, which allowed cell dissociation at a very critical time point where increased aggregation occurred. Cell dissociation presumably sorted the cells to have a better access to nutrients, such as glucose, which was also depleted sooner, compared to the VLP production process. The β-dispersion of the two systems was highly comparable at the time point of the lPEI-mediated transient transfection, and shortly thereafter. However, other courses differed significantly from each other. Therefore, a direct comparison is very difficult.

Using dielectric spectroscopy, we were able to monitor the lPEI-mediated transient transfection process inline. The DS thereby offers the chance to gain more insight in all relevant process steps of a transient transfection, e.g., the uptake of the polyplex and the formation of the VLP. It is conceivable that transfection processes with cationic polymers can be better controlled.

### 4.4. Characterization of the YF-VLP

Using the Western blot with the specific prM protein antibody, we confirmed the presence of the precursor prM protein (~22 kDa) and the membrane protein M (~8 kDa), cleaved by furin protease, in the supernatant, which has also been detected by other groups [97,98,99,100]. The semi-quantitative analysis, referring to the first M band (26 hpt) in the WB, showed that the concentration increased from 26 hpt to harvest at 96 hpt by a factor of 6. A possible reason for the presence of prM protein in the supernatant may be the inefficient maturation process during the life cycle of the VLP, or cell lysis. Additionally, mature, partially immature, or even completely immature VLPs are formed, in which the prM might be present. Although we were able to show by semiquantitative analysis that the prM protein content in the supernatant increased over time, we cannot distinguish between secreted mature YF-VLP or released immature YF-VLP and VLP proteins from damaged cells from these data.

We also detected protein-M-positive particles with two different methods (IFCM, TEM). IFCM measurements showed approximately 2.23 × 10^7^ ± 0.43 × 10^7^ labelled particles/mL, using a fluorochrome-conjugated prM antibody. The advantage of using IFCM is that the YF-VLP does not need to be linked to reporter proteins such as eGFP. Thus, the production of the YF-VLP is sufficient here and the IFCM enables a highly specific detection. The YF-VLP concentration is low compared to, e.g., González-Domínguez et al. [19]. They produced Gag VLPs and measured a particle concentration of 2.2 × 10^9^ ± 0.8 × 10^9^ particles/mL after purification, using Nanoparticle Tracking Analysis (NTA). González-Domínguez et al. used traditional NTA for a Gag-eGFP VLPs concentration determination. However, NTA measurements are significantly more non-specific in comparison to IFCM with regard to extracellular particles (EVs, including exosomes), protein aggregates, viruses, or liposomes of a certain size. Droste et al. [50] were able to show that the results of the two methods can differ significantly, since NTA measures significantly more non-specific particles [101].

Higher concentrations of YF-VLP can also achieved by an intensified downstream procedure. A 20–30% (*w/v*) sucrose density gradient can be used with an ultracentrifuge to further concentrate the VLPs, which has been successfully used [102,103]. Tangential flow filtration (TFF) [104,105] or affinity chromatography, using fast protein liquid chromatography (FPLC) [106], offers a scalable, and therefore more efficient method in the long run.

We expected a flavivirus VLP size of about 20–200 nm [21,97,101,107,108]. In the TEM, we detected YF-VLP size of 66.0 ± 17.4 nm (n = 93), which was within the known range.

## 5. Conclusions

To our knowledge, this is the first production of Yellow Fever VLP via an lPEI-mediated transient transfection process in a stirred-tank bioreactor with HEK 293T/17 SF cells, using dielectric spectroscopy as an inline PAT tool. The overall objective of this work is to gain process understanding in the production of VLPs in order to respond to global outbreaks of neglected tropical diseases through passive immunization on the platform of VLPs. In this work, we optimized the pDNA production via a statistical experimental design, and optimized the transfection-grade pDNA yield to 11 mg/L in shake flasks, which was sufficient for a transient transfection on a bioreactor scale. The final medium composition was 10 g/L glucose, 10 g/L NaCl, and 10 g/L yeast extract, with a 150 mM Na_2_HPO_4_/KH_2_PO_4_ buffer. We transferred the transient transfection process from shake flasks to the bioreactor scale. For this purpose, we investigated several process parameters, e.g., polyplexation time, cell density, medium exchange, influence of Pluronic F-68, and the influence of the anti-clumping agent on the transfection efficiency. We finally ended with an intensified process set-up with a reduced polyplexation time, a low cell density, and no medium exchange to obtain a high transfection efficiency. The implementation of dielectric spectroscopy gave deeper insights and process understanding in the VLP production process on the bioreactor scale. We correlated the permittivity signal to the viable biomass, and could monitor the uptake of the polyplex by changes in permittivity and *f_C_*. We found that the signal of *f_C_*, Δ*ε* and *α* changed, depending on the product (intracellular GFP vs. extracellular VLP), but that the plot of Δ*ε* against *ε* was very characteristic for the transient transfection process in general. Dielectric spectroscopy and β-dispersion can be used to identify the correct time point for transient transfection throughout the cell cycle. As mentioned at the beginning, the cells in the G2/M phase are easier to transfect because of the breakdown of the membrane. We confirmed the presence of YF-VLP-protein prM and M via WB, a VLP concentration of 2.23 × 10^7^ ± 0.43 × 10^7^ labelled particles/mL via IFCM, and the size of the YF-VLPs of 66.0 ± 17.4 nm via TEM. Nevertheless, the process still requires further development, e.g., as the fed-batch mode or even perfusion can provide higher yields of YF-VLPs [21]. Major problems are caused by cell aggregation, especially immediately after an lPEI-mediated transient transfection. With Pluronic F68 and a targeted use of ACA, we were able to counteract this to some extent, but did not solve it completely.

The application of cationic pH-responsive polymers, the polypiperazines, is also promising. These avoid the cytotoxic effect of lPEI and represent a new hydrophilic class of pH-responsive polymers, whose response can be tuned within the relevant pH 5–7.4 with high transfection efficiencies [109].

Our YF-VLP production process can be used as a platform. The plasmid for transient transfection can be easily replaced by a plasmid encoding proteins from other viruses. Thus, VLPs for a variety of neglected tropical disease can be produced, while the transient production process itself remains largely unchanged.

## Figures and Tables

**Figure 1 viruses-15-02013-f001:**
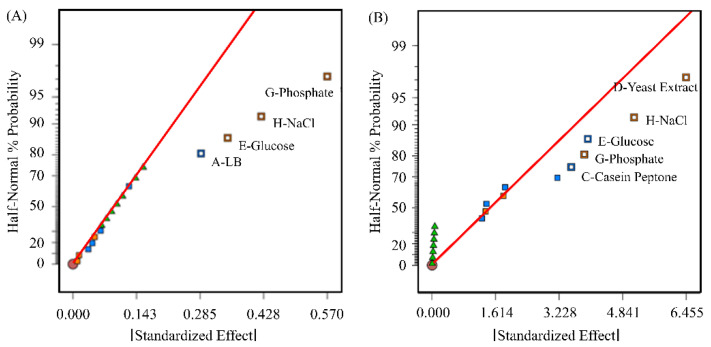
Results of the Plackett–Burman design for the *E. coli* medium optimization. (**A**) Half-normal plot of effects in relation to the measured specific pDNA yield in RFU/ODV. (**B**) Half-normal plot of effects in relation to the measured maximum OD_600_. Positive effects are shown in orange, negative effects in blue. The green triangles represent the test series of the center point. The red straight line is the cumulative probability function of a normal distribution.

**Figure 2 viruses-15-02013-f002:**
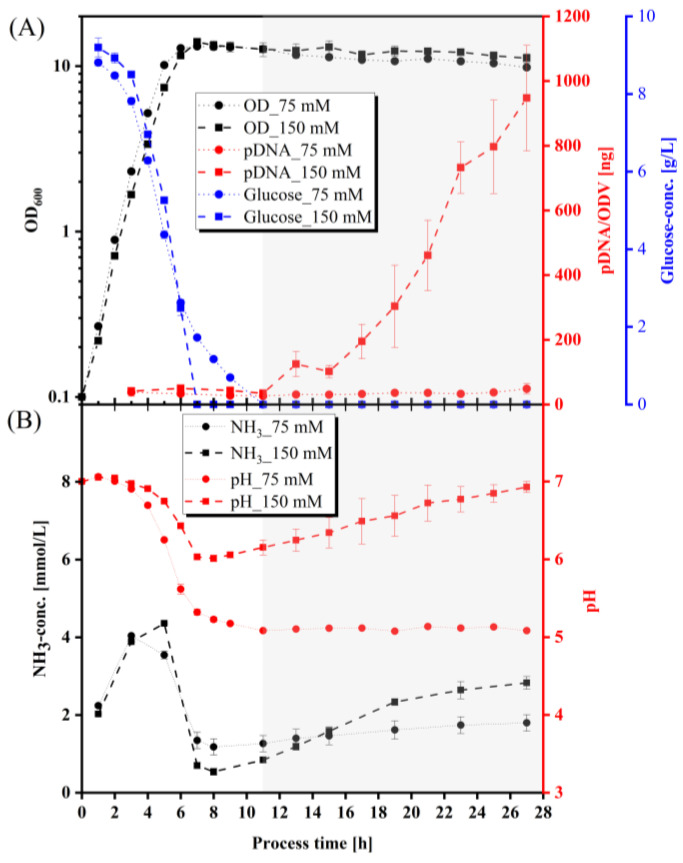
Kinetics under different Na_2_HPO_4_/KH_2_PO_4_ phosphate buffer capacities of 75 mM and 150 mM. (**A**) Semi-logarithmic growth kinetics of *E. coli*, specific pDNA production kinetics, glucose progression; (**B**) NH_3_ formation and pH value (n = 3). Initial pH value was 7.0. Circle and continuous line: 75 mM phosphate concentration. Square and dashed line: 150 mM phosphate concentration.

**Figure 3 viruses-15-02013-f003:**
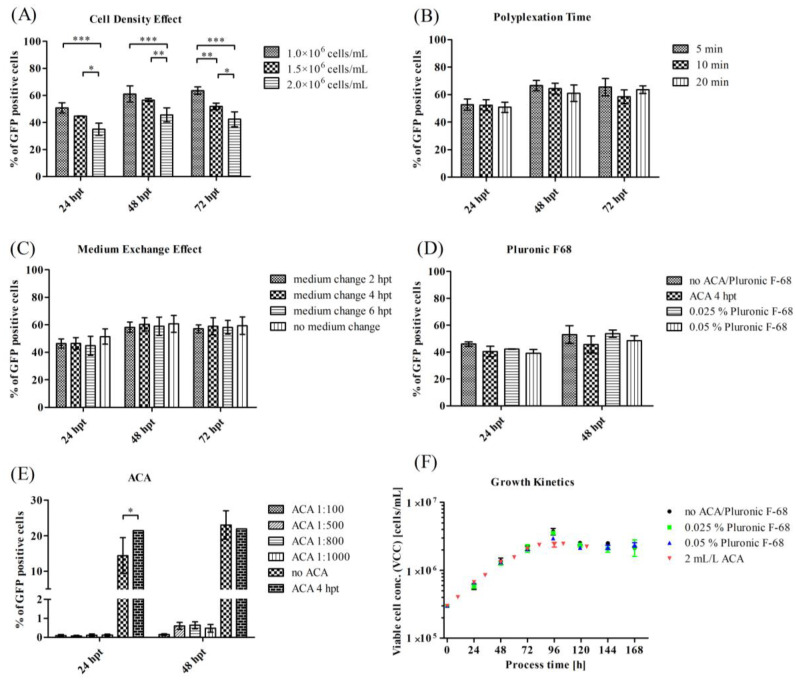
Influence of different process parameters on the transfection efficiency of HEK 293T/17 SF cells, using the model plasmid pcDNA 3.1(+)-JEV-E-EGFP in shake flasks. Transfection concentrations: 1.1 pg pDNA/cell and 6.6 pg lPEI/cell. Cells were transfected at a concentration of 1 × 10^6^ cells/mL, unless stated otherwise. Influence of (**A**) cell density (full data in Figure A3 in the Appendix A), (**B**) polyplex formation time (full data in Figure A4 in the Appendix A), (**C**) medium exchange (full data in Figure A5 in the Appendix A), (**D**) Pluronic F-68, and (**E**) anti-clumping agent on transfection efficiency of HEK 293T/17 SF cells. (**F**) Growth kinetics in shake flasks with a potential bioreactor setting of different Pluronic F-68 concentrations and ACA. * *p* < 0.05, ** *p* < 0.01, *** *p* < 0.001.

**Figure 4 viruses-15-02013-f004:**
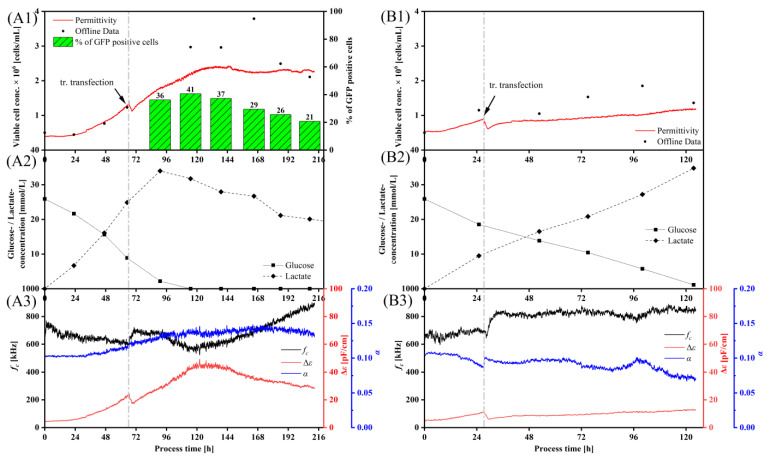
Transient transfection process data. Growth curves in correlation to the permittivity measurements and the transient transfection of HEK 293T/17 SF cells in a bioreactor, running in batch mode. (**A1**–**A3**): Transient transfection with the model plasmid pcDNA 3.1(+)-JEV-E-EGFP. (**B1**–**B3**): With the Yellow-Fever-VLP-plasmid pcDNA 3.1(+)-JEV-prM-E. (**A1**,**B1**): Offline measurement of cell concentration and calculated cell concentration from permittivity signal, including transfection efficiency data. (**A2**,**B2**): Metabolic data from batch mode—glucose and lactate measurements. (**A3**,**B3**): Measured β-dispersion data from transient transfection runs. Vertical dashed grey line in the graphs indicate the timepoint of transient transfection.

**Figure 5 viruses-15-02013-f005:**
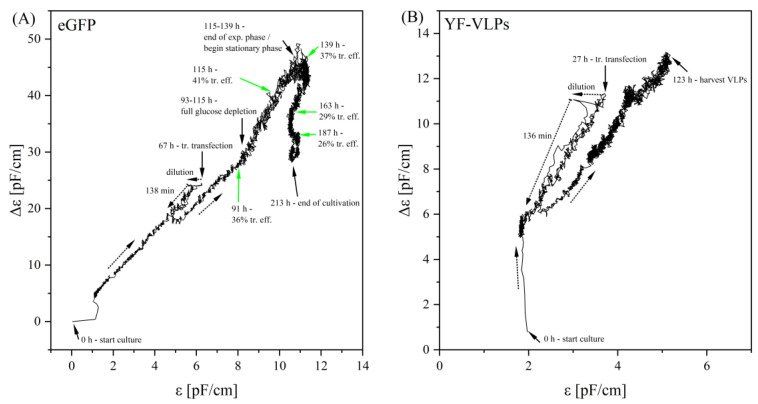
Δ*ε*-*ε*-plots of transient transfection processes. (**A**) eGFP: transient transfection with the model plasmid pcDNA 3.1(+)-JEV-E-EGFP, and (**B**) YF-VLPs: with the Yellow-Fever-VLP-plasmid pcDNA 3.1(+)-JEV-prM-E. The arrows indicate either the direction of the plot, or an important process phase. Plot A additionally shows the transfection efficiencies with the respective times. The signal drop in A took place after 138 min, and after 136 min in B. Dashed lines indicate the direction, solid arrows indicate an event during cultivation. The plots belong to the process data from Figure 4A,B.

**Figure 6 viruses-15-02013-f006:**
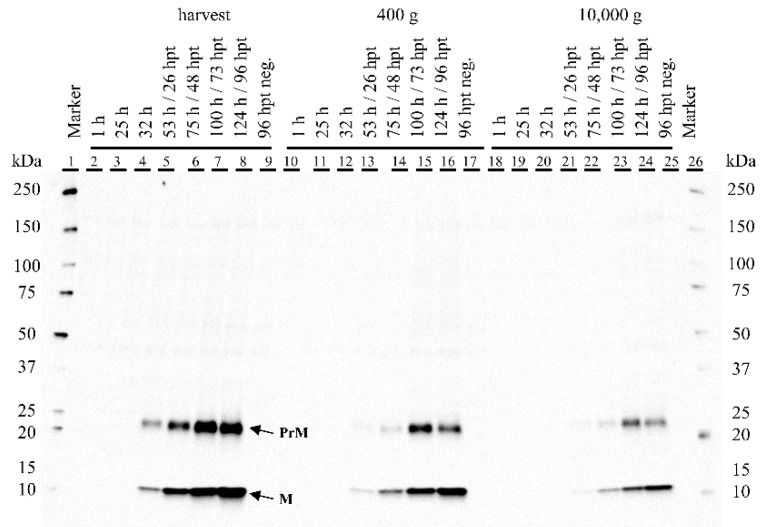
Western blot of supernatant samples of transiently transfected HEK 293T/17 SF cells with pcDNA 3.1(+)-JEV-M-E in the bioreactor. Marker lane 1 and 26: Precision Plus Protein™ WesternC (Bio-Rad, Feldkirchen, Germany). The samples were centrifuged as described in Section 2.3.7. Lane 2–8: Direct harvest from the bioreactor. Lane 10–17: 400× *g*. Lane 18–25: 10,000× *g*. Lane 9, 17 and 25: Negative control. Corresponding SDS-PAGE gel is shown in Figure A9 in the Appendix A. The semi-quantitative analysis of expressed protein prM and M in lane 4–8 is shown in Figure A10 in the Appendix A.

**Figure 7 viruses-15-02013-f007:**
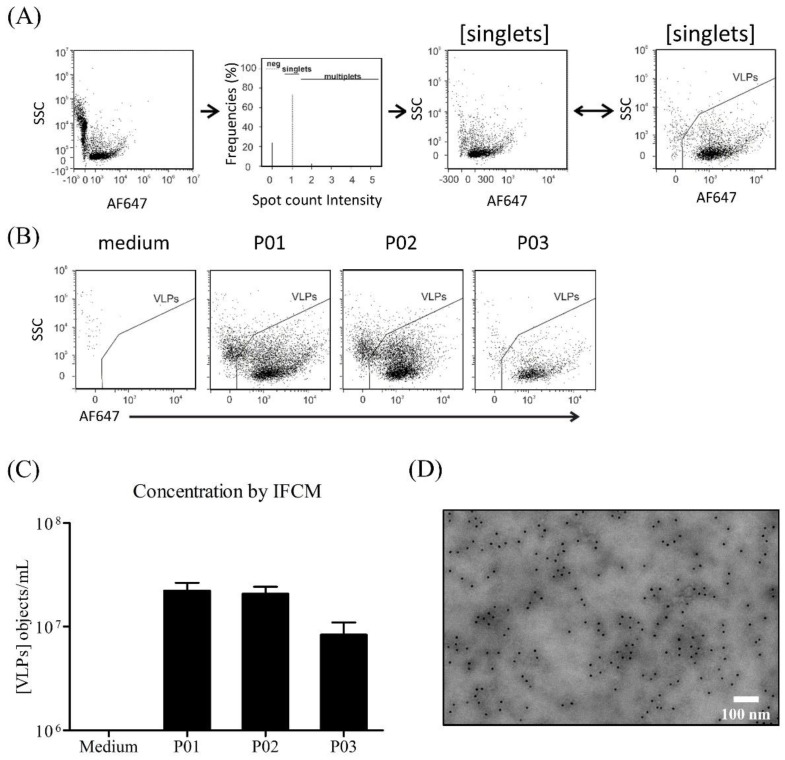
Characterization of YF-VLPs by image flow cytometry on an AMNIS StreamX (**A**–**C**) and TEM (**D**). Supernatants and medium control were analyzed by IFCM for the specific Yellow Fever virus prM protein antibody, IgG (GTX133957), conjugated with far-red-fluorescent dye Alexa Fluor 647 (AF647) (λ_ex_ 650 nm; λ_em_ 671 nm). (**A**) From all recorded signals (first plot from left), signals not showing spot counts or signal multiplets were excluded (second plot from left). The two plots on the right show side scatter (SSC) intensities of single objects plotted against the fluorescence intensities of the Yellow Fever virus prM Protein. The fourth plot is a zoomed version of the third plot. (**C**) Results of four different samples are given in objects/mL for the Yellow Fever virus prM Protein^+^ VLPs (mean ± SD). P01: Supernatant of reactor batch_10, lane 8 in Figure 6 (YF-VLP); P02: Clarified supernatant (10 min, 400× *g*), lane 16 in Figure 6; P03: Clarified supernatant (15 min, 10,000× *g*), lane 24 in Figure 6. (**D**) TEM picture of negative staining and immunogold labelling of YF-VLPs. Immunogold labelling with prM antibody, conjugated with gold beads of 10 nm in diameter. Gold particles bound to the surface of the YF-VLPs are seen as small black dots.

**Table 1 viruses-15-02013-t001:** Placket–Burman screening design for an optimization of E. coli growth and a specific pDNA-production (ng pDNA/ODV). The following concentrations of media components were studied in the PB design: High (coded as 1): 10 g/L (0.5 g/L for MgSO_4_); Center-point (coded as 0): 5.5 g/L (0.275 g/L for MgSO_4_); Low (coded as −1): 1.0 g/L (0.05 g/L for MgSO_4_), respectively.

Standard	8	6	5	1	7	3	4	12	11	9	10	2	Center Point							
Run	1	2	3	4	5	6	7	8	9	10	11	12	13	14	15	16	17	18	19	20
LB	1	−1	−1	1	1	1	−1	−1	1	1	−1	−1	0	0	0	0	0	0	0	0
Meat-peptone	1	−1	−1	1	−1	−1	1	−1	−1	1	1	1	0	0	0	0	0	0	0	0
Casein-peptone	−1	−1	1	−1	−1	1	−1	−1	1	1	1	1	0	0	0	0	0	0	0	0
Yeast extract	−1	1	−1	1	−1	1	1	−1	1	−1	1	−1	0	0	0	0	0	0	0	0
Glucose	−1	−1	1	1	1	−1	1	−1	1	−1	−1	1	0	0	0	0	0	0	0	0
Glycerol	1	1	1	1	−1	1	−1	−1	−1	−1	−1	1	0	0	0	0	0	0	0	0
Phosphate *	−1	1	−1	−1	1	1	1	−1	−1	1	−1	1	0	0	0	0	0	0	0	0
NaCl	1	−1	1	−1	1	1	1	−1	−1	−1	1	−1	0	0	0	0	0	0	0	0
MgSO_4_	1	1	1	−1	−1	−1	1	−1	1	1	−1	−1	0	0	0	0	0	0	0	0
Dummy 1	−1	1	1	1	1	−1	−1	−1	−1	1	1	−1	0	0	0	0	0	0	0	0
Dummy 2	1	1	−1	−1	1	−1	−1	−1	1	−1	1	1	0	0	0	0	0	0	0	0

Note: ***** Na_2_HPO_4_/KH_2_PO_4_.

## Data Availability

The data that support the findings of this study are available from the corresponding author, Denise Salzig, upon reasonable request.

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
