# Peer review of "A Bioreactor-Based Yellow Fever Virus-like Particle Production Process with Integrated Process Analytical Technology Based on Transient Transfection"

_viruses, 2023, doi:10.3390/v15102013_

Round 1

Reviewer 1 Report

In this manuscript, Dekevic and colleagues have defined parameters that allow optimized bulk production of yellow fever virus-like particles (VLPs) from transfected mammalian cells.  This is a significant line of investigation due to the potential of VLPs to act as an effective vaccine.  Procedures are optimized for large-scale production of the plasmid DNA from E. coli in addition to the transient transfection of mammalian suspension cell cultures and production of the VLPs.  VLPs produced using the optimized procedures are characterized by TEM as well as detection of prM and M proteins on blots.  Information gained from this study is potentially useful for industrial production of a wide range of VLPs in addition to yellow fever VLPs.

There are some weaknesses that should be addressed prior to publication as described below.  

1.  The version of the manuscript provided to reviewers is not referenced correctly, as almost all of the citations are replaced with internal error messages.  Figures are also not referenced correctly.  For example, see the sentence starting on Line 389: “We further reduced the polyplex formation time (Error! Reference source not found. B) from 20 min to 10 min or 5 min, and found no significant changes in the transfection efficiency.”.  Nearly every instance in the manuscript where a citation is supposed to be has this error in its place.  Nearly every instance in the manuscript where a figure is referenced, a garbled version appears in its place (in this case the “B)” presumably is meant to refer to Fig. 3B).  Combined together, these issues make the manuscript exceptionally difficult to read and evaluate.  Extensive proofreading is needed to fix all of these problems.  

2.  Resolution of Fig. 1 is poor, making text difficult to read.  Figs. 3 and 6 could also be improved in this respect, although they are in much better shape than Fig. 1.

3.  Authors should elaborate on the decision-making process in their discussion of Fig. 3A.  Is the goal to maximize the fraction of cells that are GFP-positive, or is the goal to maximize the absolute number of cells that are GFP-positive?  If it is the fraction that are positive, please explain the reasoning as it relates to industrial production of VLPs.  If it is the absolute number of cells, then the discussion and data analysis with respect to cell density should reflect this.

4.  Fig. 3D is lacking proper controls. Why is the control “no ACA” when the main variable being investigated is Pluronic F68?

5.  Fig. 4A1 indicates that transfection efficiency is changing along a timecourse.  Is the fraction of GFP-positive cells really changing due to differences in transfection efficiency as indicated?  Or is it changing due to differential growth rates of the two cell populations following transfection?  I would recommend labeling this as GFP-positive cells rather than as Transfection Efficiency.

6.  Figs. S9 and S10 should be combined.  It is not clear which of the many bands apparent on the blot in S9 are included in the quantification shown in S10.  

7.  Fig. 6.  The 400g and 10,000g labels suggest that there was some kind of centrifugation, but this is not described.  It is not clear if we are looking at pellet or supernatant fractions.  How much of the prM and M signals shown in Fig. 6 (and quantified in Fig. S10) actually comes from VLPs as opposed to leakage of unassembled proteins out of damaged cells?  

Minor proofreading needed.

Reviewer 2 Report

The aim of this study was to develop a process for production of Yellow Fever Virus-like Particles (VLPs) based on transient transfection in a stirred-tank bioreactor. Design of experiments, combined with novel measurements, including integrated dielectric spectroscopy (for viable cell densities) and imaging flow cytometry (for VLP measures), contribute to a useful potential platform for VLPs and other immunizing agents. Detailed comments follow:

1) Line 94. The empirical parameter, alpha, is mentioned, but it is not clear where it appears in Equations (1, 2, or 3).

2) Multiple locations in the manuscript "Error! Reference source not found" (lines 167, 333, 349, 389, 390, 395...) suggest this submitted manuscript was not carefully checked before submission.

Reviewer 3 Report

The manuscript by Dekevic et al attempts to explore the bioreactor-based YF-VLP production process with integrated process analytical technology based on transient transfection. The primary focus of the research was to achieve the production of well-formed VLPs, which required the thorough exploration and optimization of parameters from plasmid DNA production to the bioreactor stage. However, this work only made limited progress comparing to shaking flask because fed-batch or perfusion culture were not applicated in this work. Thus it hardly can offer helpful experiences for practical industrial production. And a comparative analysis between VLPs produced via bioreactor and shaking flask is necessary to address the advantages of this work. In short, this work failed to delineate the distinct advantages inherent to bioreactor-based VLP production adequately. Furthermore, in this study, only TEM was used for the characterization of VLP. It would be better if the more aspects of VLP could be considered, such as the particle size, heterogeneity, sedmentation coefficient, biological activity, and immunogenicity of VLP produced from different processes. More detailed comments are offered below:

Major comments:

1.      The entire article does not mention the corresponding graph when describing the result, which is not standard.

2.      Line 347, whether this yield results from a single experiment or multiple repetitions?

3.      Line 395-396, this result is a little confusing as to whether this step is necessary, and medium exchange generally considered after several hours of transfection.

4.      Line 400-402, the description was not completely consistent with the graph, because the addition of Pluronic F68 in different concentrations had little effect on the transfection efficiency.

5.      Line 403, line 415, in addition to ensuring transfection efficiency, it is also necessary to consider the concern of cell aggregation during the culture process, and it needs to consider whether to add ACA and at what point.

6.      Line 432, firstly, the literature source should be noted, and then, can the highest titer be maintained 24-96 h after transient transfection?

7.      Line 434-435, this conclusion requires a comparison of transfection efficiency at multiple time? points.

8.      Line 495-506, the description of the results is not clear, why not correlate Δε with transfection efficiency and cell state throughout the process?

9.      Line 528-529, how to explain the result of Western blotting for protein E, and why not use specific antibody tests?

10.  Figure 3F, Figure 4, the unit of cell density on the vertical is irregular,

11.  Figure 7A, 7B, the horizontal and vertical coordinates were not clear, it is hard to understand.

12.  There are many grammatical and syntactic errors in the manuscript, which need a proofread throughout the manuscript.

Minor comments:

1.      Line 335, line 365, “E. coli” should be “E. coli

2.      Line 432, “in” should be “before”

3.      Line 463, a logical error, “Consequently” should be changed to “then” or “subsequently”.

4.      Line 470, line 633, a grammatical error, “cell densities” should be “cell density”.

5.      Line 476, a space is required at the beginning of the paragraph.

6.      Line 481-482, a grammatical error, “different compared to” should be “different from”.

7.      Line 503, a grammatical error, “and by a higher signal noise” should be “and a higher signal noise”.

8.      Line 504, a logical error, “Here” should be “Therefore”.

9.      Line 519-522, it might be more appropriate to change the sentence in line 519-522 to be more clear and logical.

10.  Some figures are out of order, should be added A, B, C……

11.  Line 538, line 540, line 542, there are problems with the statement logic.

12.  Line 574, a grammatical error, “val-uable” should be “valuable”

13.  Line 584-585, a logical error, “respectively” should be “and”.

14.  Line 587-590, line 591-592, line 628-630, it might be more appropriate to change the sentence in line 587-590, line 591-592 and line 628-630 to be more clear and logical.

15.  Line 612, a logical error, “and” should be deleted.

16.  Line 624, a grammatical error, “little to no” should be “little or no”.

17.  Line 631, “impact” and “effect” are used repeatedly.

18.  Line 637-638, a grammatical error, the last two “by” should be deleted.

19.  Line 645, a grammatical error, “avoid/minimize” should be “avoid or minimize”.

    There are many grammatical and syntactic errors in the manuscript, which need a proofread throughout the manuscript.

Author Response

We would like to thank reviewer 3 for the time and effort they invested in reading and commenting on our manuscript, and we are sure that the quality of the publication has improved by incorporating his/her recommendations.

We addressed each comment below. Please note that his may have changed the page number, a figure number or the line in the manuscript.

Major comments:

  1. The entire article does not mention the corresponding graph when describing the result, which is not standard.

Authors: We checked the whole manuscript again. Every single illustration was mentioned in the text. Every figure in the supplement was also included in the main part of the manuscript. We suspect that errors occurred during upload that caused the links to display incorrectly. We have checked everything again and corrected it.

  1. Line 347, whether this yield results from a single experiment or multiple repetitions?

Authors: The result came from a triple determination (n=3). This was described in the caption of figure 2. However, we have also included it in the text.

  1. Line 395-396, this result is a little confusing as to whether this step is necessary, and medium exchange generally considered after several hours of transfection.

Authors: A medium exchange may be useful in lPEI-mediated transient transfection because of the cytotoxic effect of lPEI. When medium is exchanged, excess lPEI is removed from the culture and the damaging effect is minimized. We added this to the manuscript.

  1. Line 400-402, the description was not completely consistent with the graph, because the addition of Pluronic F68 in different concentrations had little effect on the transfection efficiency.

Authors: Thank you for the comment. A two-way analysis of variance (ANOVA) was performed to determine the statistical significance among different parameters. Statistical differences were defined as follows: *p < 0.05, **p < 0.01, ***p < 0.001. The difference in transfection efficiencies using different concentrations of F-68 is not statistically significant and was not interpreted as such.

  1. Line 403, line 415, in addition to ensuring transfection efficiency, it is also necessary to consider the concern of cell aggregation during the culture process, and it needs to consider whether to add ACA and at what point.

Authors: Thank you for this comment. Indeed, the timing of the addition of ACA is extremely crucial. On the one hand, it was important to wash the ACA out of the culture 24 h before transient transfection and on the other hand, ACA could not be added to the culture earlier than 4 h after transient transfection because it inhibits the transient transfection process up to 100%. Cell aggregates formed mainly after lPEI-mediated transient transfection and therefore we decided to add ACA after the uptake of the polyplexes was complete (earliest 4 hpt).

  1. Line 432, firstly, the literature source should be noted, and then, can the highest titer be maintained 24-96 h after transient transfection?

Authors: Thank you for your comment. We have added the literature. In a transient transfection process, the highest titer will quickly drop off again, as the transfected pDNA is excreted by the cell within a few days. To maintain the titer, a stably transfected cell line would be needed that maintains the VLP titer over a longer period of time.

  1. Line 434-435, this conclusion requires a comparison of transfection efficiency at multiple time?

Authors: We agree with your statement. The 41% transfection efficiency 48 hpt was related to Figure 4A1. We have added this to the manuscript.

  1. Line 495-506, the description of the results is not clear, why not correlate Δε with transfection efficiency and cell state throughout the process?

Authors: The advantage of the β-dispersion over the Δε-ε-plot is that important phases or events during a cultivation can be detected, such as physiological changes of the cell. For example, the actual transfection process can be monitored, a nutrient deficiency or the transition into the stationary phase or the initiation of the apoptosis.

  1. Line 528-529, how to explain the result of Western blotting for protein E, and why not use specific antibody tests?

Authors: Thank you for this comment. The antibody against protein E in the Western blot bound very nonspecifically. On the one hand, we found these non-specific bindings in the suspension system, but also in the adherent system where FBS was used. Here, there was also a lot of non-specific binding with the FBS. In the suspension system we assume that this is a non-specific binding of the antibody with components of the HEK cell.

We have only used protein E-specific monoclonal antibodies. Since it was misleadingly expressed in the manuscript, we have modified the sentence.

  1. Figure 3F, Figure 4, the unit of cell density on the vertical is irregular,

Authors: We have modified the unit of cell density.

  1. Figure 7A, 7B, the horizontal and vertical coordinates were not clear, it is hard to understand.

Authors: We added the description for AF647 (Alexa Fluor 647), a far-red-fluorescent dye and SSC (side scatter) in the figure caption.

  1. There are many grammatical and syntactic errors in the manuscript, which need a proofread throughout the manuscript.

Authors: Thank you for this comment. During the preparation of the manuscript, strict attention was paid to ensure that the links and references are correct. This was checked several times and everything was linked correctly. Unfortunately, there must have been a problem uploading the manuscript that caused this issue. We have revised the manuscript again and strictly made sure that the references are displayed correctly. The proofreading was done by a native speaker. 

Authors: The following minor comments were summarized and edited in the manuscript.

  1. Line 335, line 365, “E. coli” should be “E. coli”
  2. Line 432, “in” should be “before”
  3. Line 463, a logical error, “Consequently” should be changed to “then” or “subsequently”.
  4. Line 470, line 633, a grammatical error, “cell densities” should be “cell density”.
  5. Line 476, a space is required at the beginning of the paragraph.
  6. Line 481-482, a grammatical error, “different compared to” should be “different from”.
  7. Line 503, a grammatical error, “and by a higher signal noise” should be “and a higher signal noise”.
  8. Line 504, a logical error, “Here” should be “Therefore”.
  9. Line 519-522, it might be more appropriate to change the sentence in line 519-522 to be more clear and logical.
  10. Some figures are out of order, should be added A, B, C……
  11. Line 538, line 540, line 542, there are problems with the statement logic.
  12. Line 574, a grammatical error, “val-uable” should be “valuable”
  13. Line 584-585, a logical error, “respectively” should be “and”.
  14. Line 587-590, line 591-592, line 628-630, it might be more appropriate to change the sentence in line 587-590, line 591-592 and line 628-630 to be more clear and logical.
  15. Line 612, a logical error, “and” should be deleted.
  16. Line 624, a grammatical error, “little to no” should be “little or no”.
  17. Line 631, “impact” and “effect” are used repeatedly.
  18. Line 637-638, a grammatical error, the last two “by” should be deleted.
  19. Line 645, a grammatical error, “avoid/minimize” should be “avoid or minimize”.

Round 2

Reviewer 1 Report

The revised manuscript has addressed my concerns.  Authors acknowledge in their response to concern #7 that to some extent the signals detected in Fig. 6 could be influenced by leakage of prM and M from damaged cells.  This should also be acknowledged in the manuscript so that readers can have a better sense of the strengths and limitations of this particular experiment.  I have no additional concerns regarding the revised version of this manuscript. 

Reviewer 3 Report

According to the previous review comments, the author has responded to and revised most of them, but there are still some areas that have not been revised. In addition, the previous question of adequate characterization of VLP was not answered. It would be better if the more aspects of VLP could be considered, such as the particle size, heterogeneity, sedimentation coefficient, biological activity, and immunogenicity of VLP produced from different processes. More detailed comments are offered below:

Minor comments:

1.      All figures should be marked A, B, C, etc.

2.      The value “1005 , 1006, 1007” of Figure 3F should be “105, 106, 107”.

3.      The A, B in Figure 5 should be outside the diagram.

4.      As shown in Figure 7D, the particle morphology of 200 nm bar is not clear, so it is recommended to replace it with 100 nm bar or smaller bar.

5.      Please note the table specification and revise it.
